# Phage-Encoded Endolysins

**DOI:** 10.3390/antibiotics10020124

**Published:** 2021-01-28

**Authors:** Fatma Abdelrahman, Maheswaran Easwaran, Oluwasegun I. Daramola, Samar Ragab, Stephanie Lynch, Tolulope J. Oduselu, Fazal Mehmood Khan, Akomolafe Ayobami, Fazal Adnan, Eduard Torrents, Swapnil Sanmukh, Ayman El-Shibiny

**Affiliations:** 1Center for Microbiology and Phage Therapy, Biomedical Sciences, Zewail City of Science and Technology, Giza 12578, Egypt; fabdelrahman@zewailcity.edu.eg (F.A.); sragab@zewailcity.edu.eg (S.R.); 2Department of Biomedical Engineering, Sethu Institute of Technology, Tamil Nadu 626115, India; maheswaran.easwaran@gmail.com; 3Department of Biomedical Laboratory Science, College of Medicine, University of Ibadan, Ibadan 200284, Nigeria; daramolaisaac6716@gmail.com (O.I.D.); johntolu98@gmail.com (T.J.O.); ayobamiakomolafe@gmail.com (A.A.); 4School of Life Sciences, La Trobe University, Melbourne, VIC 3086, Australia; Stephanie.Lynch@latrobe.edu.au; 5Center for Biosafety Mega-Science, Key Laboratory of Special Pathogens and Biosafety, Wuhan Institute of Virology, Chinese Academy of Sciences, Wuhan 430071, China; fmkhanmicrobiologist@mails.ucas.ac.cn; 6International College, University of Chinese Academy of Sciences, Beijing 100049, China; 7Atta ur Rahman School of Applied Biosciences (ASAB), National University of Sciences and Technology (NUST), Islamabad 24090, Pakistan; adnanfazal@asab.nust.edu.pk; 8Bacterial Infections: Antimicrobial Therapies Group, Institute for Bioengineering of Catalonia (IBEC), The Barcelona Institute of Science and Technology (BIST), 08028 Barcelona, Spain; etorrents@ibecbarcelona.eu; 9Microbiology Section, Department of Genetics, Microbiology, and Statistics, Faculty of Biology, University of Barcelona, 08028 Barcelona, Spain

**Keywords:** endolysin, antibiotic resistance, bacteriophages

## Abstract

Due to the global emergence of antibiotic resistance, there has been an increase in research surrounding endolysins as an alternative therapeutic. Endolysins are phage-encoded enzymes, utilized by mature phage virions to hydrolyze the cell wall from within. There is significant evidence that proves the ability of endolysins to degrade the peptidoglycan externally without the assistance of phage. Thus, their incorporation in therapeutic strategies has opened new options for therapeutic application against bacterial infections in the human and veterinary sectors, as well as within the agricultural and biotechnology sectors. While endolysins show promising results within the laboratory, it is important to document their resistance, safety, and immunogenicity for *in-vivo* application. This review aims to provide new insights into the synergy between endolysins and antibiotics, as well as the formulation of endolysins. Thus, it provides crucial information for clinical trials involving endolysins.

## 1. Introduction

Endolysins are gaining importance in recent years due to their broad lytic activities against Gram-positive and Gram-negative bacterial cells [1,2]. Endolysins are bacteriophage-encoded enzymes, which act by hydrolyzing the host cell wall and subsequently allowing the release of bacteriophage progenies. Therefore, such enzymes are essential components of the lytic phage life cycle and are a promising alternative to antibiotics [1,2]. The lytic activity of endolysins is classified into different types, namely; (a) acetylmuramidases, (b) transglycosylases, (c) glucosaminidases, (d) amidases, and (e) endopeptidases [1,3]. These five different types are further explained in this review to emphasize their mode of action, as well as regulation of expression. As mentioned, endolysins are involved in peptidoglycan degradation during cell lysis, which is regulated by different phage lytic enzymes and holins [4]. Holins are small hydrophobic hole-forming proteins (>100 nm in diameter) [4,5,6,7,8]. Notably, besides a few exceptions such as lysozyme or lysostaphin, phage proteins rarely develop resistance among their bacterial hosts, mostly due to the horizontal gene transfer among phage–host systems over a long time [1]. However, bacterial hosts additionally develop resistance mechanisms against free endolysins such as development of the outer membrane (capsule), exopolysaccharides (biofilms), etc. In a similar realm, the host immune system such as pro-inflammatory cytokines and antibodies may have implications in the use of endolysin; both aspects will be discussed in further detail throughout this review.

The research surrounding phage lytic enzymes as an alternative for antibiotic resistance has rapidly increased. However, due to the lack of *in-vivo* studies and sufficient clinical trials, the use of bacteriophages and endolysins for the development of an effective phage therapy has been hindered [3]. Considering the recent progress in this field, our review aims to corroborate the research and applicability of phage endolysins for *in-vivo* therapies. It aims to provide information to continue advancing endolysin research. It also highlights the importance of phage lytic enzymes as an effective alternative against antibiotic-resistant pathogens and discusses their challenges and limitations.

## 2. History of Endolysins

The knowledge that phage lysates contain enzymatic activity that could cause in vitro lysis of bacteria was initially documented by Frederick W. Twort during his discovery of bacteriophages [9]. Twort noted in his seminal manuscript what was perhaps the first evidence of endolysins, stating that there appeared to be a non-transmissible, heat-labile property that produced transparent zones of lysis [9]. We now attribute this to either endolysins or virion-associated peptidoglycan hydrolases (VAPGHs) of the staphylococcal phage [10]. Importantly, endolysins, are different from virion-associated peptidoglycan hydrolases (VAPGHs) as endolysins are secreted at the end of the phage lytic cycle [2]. In contrast, VAPGHs are released from phage tail tips at the initial stage of peptidoglycan penetration [3]. Twort later defined a “transmissible virus” (bacteriophages) as only acting on live bacteria, whereas a non-transmissible “bacterial lysin” secreted by the virus would act on dead bacteria [11]. To accompany this theory, dead staphylococci could not be lysed by phage unless a small amount of live staphylococci was added, which released some emerging agents [11]. By 1926, phage biologist F. D. Reynals confirmed Twort’s lytic findings by performing identical experiments on both Gram-positive and Gram-negative species, and noting that it was specific only to Gram-positive species [12].

By 1934, Alice Evans proceeded to classify bacterial strains using phage, founding the analytical field of phage typing [13]. W. R. Maxted later launched an investigation on the cause of nascent lysis reported by Evans and obtained a lytic factor from the phage filtrates to establish its origin and role in the phage lytic system [14]. By 1957, Richard Krause renamed the Evans B563 phage to C_1_ due to its defined specificity for Group C streptococci. Krause became the first scientist to partially purify the C_1_ lysin [1]. Vincent Fischetti, in 1971, prepared a highly purified C_1_ lysin as a resolution of his thesis work at the McCarthy laboratory and this allowed for further detailed studies on how surface proteins of Gram-positive organisms bind to the cell wall [1].

The term “Endolysin” was later published to represent “any one of several unrelated types of enzymes (i.e., muramidase, amidase, or transglycosylase), which attack either the glycosidic bonds (i.e., muramidase and transglycosylases) or peptide bonds (i.e., amidases) that confer mechanical rigidity on the peptidoglycan” [15]. The use of endolysins against *in-vivo* bacterial infections was later implemented in the early 2000s. Nelson et al. published the first report investigating the prophylactic use of endolysin C1 in an *in-vivo* model against upper respiratory group A streptococci [16]. Subsequently, Loeffler et al. published a novel article on the use of endolysin Cp-1 against pneumococcal bacteremia through intravenous administration [16,17]. By 2013, clinical trials for endolysin use were approved, including a phase I study to evaluate the safety of pharmacokinetics and pharmacodynamics of endolysin SAL-1 based drugs, designed to treat antibiotic-resistant staphylococcal infections [18], and a later phase I/II study using engineered chimeric endolysin Staphefekt (Micreos Human Health BV, The Netherlands)to target *Staphylococcus aureus* responsible for atopic dermatitis (Staphefekt™, developed by Micreos, is the world’s first endolysin available for human use on intact skin) [19,20] (Figure 1).

## 3. Endolysin Structure and Mode of Action

The structure of endolysins is a factor determined by their origin. However, the majority of endolysins (usually a molecular weight of 15–40 kDa) have a modular configuration, while endolysins acting on Gram-negative bacteria have a simple globular configuration [3]. Modular endolysins are often characterized by the presence of one or two (multi-domain) N-terminal enzymatically active domains (EADs) linked by a short, flexible linker region to a C-terminal cell wall-binding domain (CBD). This configuration is common to Gram-positive phages and mycobacteriophages [3]. The N-terminal enzymatically active domain (EAD) of modular endolysins functions to cleave various specific peptidoglycan bonds in the murein layer of the host bacterium, while the C-terminal cell wall-binding domain (CBD) recognizes and binds to different epitopes in the cell wall for proper fixation of the catalytic effect of the EAD [21]. Endolysins from Gram-positive phages structurally resemble fungal cellulases which are similarly constructed enzymes with EADs and CBD joined by a flexible linker [22] (Figure 2a–c).

Endolysins of phages infecting Gram-negative host can be structured in various forms; however, most are configured to have a simple globular module of the EAD (molecular weight, 15–20 kDa) without a CBD [23] (Figure 2d,e). Recent studies also show the occurrence of Gram-negative phage endolysins with globular configuration, having one or two CBDs at the N-terminal while the EAD module is toward the C-terminal [23]. Thus, there is an inverted orientation of the general organization of common endolysins from phages infecting Gram-positive hosts. There are also signal-arrest-release (SAR) Gram-negative phage endolysins which are independent of holin-mediated endolysins [24]. SAR endolysins are initially localized to the periplasmic membrane before further release from the membrane to initiate host lysis. Endolysins of Phage P1 and ERA103, Lyz_P1_ and Lyz_103_ respectively, are common examples [24,25].

**Figure 2 antibiotics-10-00124-f002:**
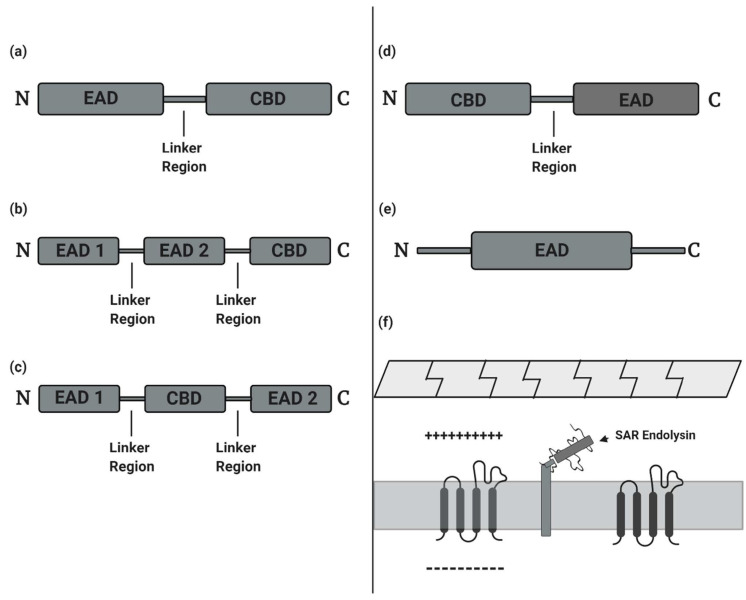
Modular configuration models of common phage endolysins. (**a**) Model with one N-terminal enzymatically active domain (EAD) and a C-terminal cell wall-binding domain (CBD). (**b**) Multi-domain model with two EADs and a C-terminal CBD. (**c**) Multi-domain model with a CBD located between two EADs. (**d**) Modular endolysin with a C-terminal EAD and an N-terminal CBD. (**e**) Simple globular model of an EAD with no CBD. (**f**) Model of a tethered signal-arrest-release (SAR) endolysin localized to the periplasmic membrane before release and activation of lysis function [24].

Based on the mode of action and individual enzymatic specificities, EADs are categorized into three classes of six distinct enzymatic activities (Figure 3):(a)Glycosidases generally cleave the β-1,4 glycosidic bonds linking alternating polymeric structures of N-acetylmuramic acids (MurNAc) and N-acetylglucosamines (GlcNAc) in the peptidoglycan layer. Subclasses of Glycosidases include N-acetyl-β-D-muramidases, which cleave bonds between MurNAc and GlcNAc; N-acetyl-β-D-glucosidases, which cleave bonds between GlcNAc and MurNAc residues; and lytic transglycosylases, which are not considered as true hydrolases as they do not require water molecules for their catalytic action [21]. Like the other two glycosidases, transglycosylases cleave β-1,4 bonds between MurNAc and GlcNAc, but also involve an intra-molecular reaction that results in the generation of a 1,6-anhydro ring at the MurNAc residue [26,27].(b)Amidases, as N-acetylmuramoyl-L-alanine amidase, catalyze the cleavage of amide bonds between the MurNAc and the first amino acid in the peptide stem moiety, L-alanine.(c)Endopeptidases cleave bonds between two amino acids of the stem peptide. Bond cleavage can either occur within interpeptide bridge or stem peptide–interpeptide bridge [27]. Examples include L-alanoyl-D-glutamate endopeptidase (VANY), c-D-glutamyl-m-diaminopimelic (DAP) acid peptidase, D-Ala-m-DAP endopeptidase, D-alanyl-glycyl endopeptidase (CHAP), etc. [28,29].

As the name suggests, the CBD binds to cell wall components such as the murein ligands or secondary cell wall polymers: teichoic acids and natural polysaccharides of the bacterium [30]. PlyG and PlyL (amidases) are endolysins binding selectively to secondary cell wall polysaccharides of the bacilli cell wall [31]. A probable hypothesis for the presence of the CBD motifs in most Gram-positive phage endolysins is that the binding of this domain to ligands of the peptidoglycan layer conserves the diffusion of endolysins and the destruction of nearby potential hosts after cell lysis is achieved, hence, allowing progeny phages to carry out novel infection stage [3]. The presence of the outer membrane barrier shielding the peptidoglycan layer of Gram-negative bacteria would discourage expulsed endolysins from gaining access to neighboring cells at the end of the lytic cycle.

To overcome the outer thick mycolate layer linked to the peptidoglycan layer in mycobacteria, the mycobacteriophages have evolved two forms of EAD lysins; Lysin A, the peptidoglycan hydrolase and Lysin B, the mycolyl arabinogalactan esterase, cleaving bonds between the mycolic acid and the arabinogalactan [28,32].

The structure of a tailed phage consists of two parts: the icosahedral head which contains and protects the phage genome and the tail, which attaches to the specific receptor site on the bacterial cell wall. Bacteriophage injects its DNA into bacteria via its tail, and a progeny phage is produced. These newly produced phages go on to lyse bacterial cells [33]. The release of phage progeny at the end of the lytic cycle is facilitated by holins and lysins. Within the bacterial host cell at the late stage of phage lytic cycle, holin is produced, which makes a hole in the bacterial cytoplasmic membrane and creates a channel to facilitate lysin delivery to the bacterial peptidoglycan, which results in cleavage of specific peptidoglycan bonds and disrupts the bacterial structure (Figure 4) [4].

Basically, there is a three-step model system of the bacterial host lysis. This model consists of three proteins named endolysin, holin, and spanin. These proteins perform actions on the bacterial outer membrane, inner membrane, and peptidoglycan and can cause the lysis of the host. There are two distinct mechanisms for the degradation of the peptidoglycan in Caudo-virales of Gram-negative hosts: (1) Holin–endolysin and (2) Pinholin SAR endolysin. In the prior step, the degradation of the peptidoglycan is initiated when holin makes small pores in the host inner membrane; then, lysis is initiated by releasing endolysin inside the periplasm and degrading the peptidoglycan. In the last step, lysis is started when the pinholin can perform the membrane depolarization which can activate the secreted SAR endolysin. Previously it was thought that for the lysis of the Gram-negative hosts the degradation of the first two barriers of the membrane is sufficient, but according to the new findings, spanin, which is another lysis protein, is required for the degradation of the outer membrane. The lysis of the bacterial host cell occurs through this mechanism [34].

Multiple *in-vivo* experiments have shown that endolysins are effective against a variety of Gram-positive bacteria such as PlyC [35], C_1_ [36], ClyR [37], Cpl-1 [38], ClyV [39], and ClyJ [40]. Phage endolysins often show robust antibacterial activity against Gram-positive bacteria but compromised activity against Gram-negative bacteria due to the presence of protective outer membrane [41,42,43]. However, in recent years molecular engineering approaches have increased the applicability of endolysins in targeting Gram-negative pathogens [44].

The peptidoglycan disruption by endolysins results in destruction of bacterial cells and the cell hydrolyzing phenomenon can be detected by a variety of enzyme assays. The enzyme activity can be detected based on clearing of bacterial suspension, native peptidoglycan degradation, a formation of a clear zone within the peptidoglycan matrix or bacteria, and a decrease in viable cell count in terms of cell forming units [45]. The endolysin enzymatic activity is commonly quantified in presence of tris or phosphate buffers, which provide an optimal environment for its lytic activity. The enzyme activity can be monitored by observing a corresponding reduction in the optical density of bacterial suspension at 600 nm at 37 °C [17,46]. Zymogram or overlay assay is a deviation of turbidity reduction assay, in which the peptidoglycan of bacteria mixed with a semi-solid matrix is cleared as compared to the liquid bacterial suspension. Overlay assays can be used for the purpose of screening large endolysin-expressing libraries while zymograms show the hydrolytic activity of endolysins on SDS-PAGE at a specific spot, which corresponds to their molecular weight [47,48,49]. The reduction in the colony-forming unit (CFU) is another commonly used method for quantifying bacterial reduction over a time of endolysin treatment [50]. For visible enzyme activity, spot assay is commonly practiced in which lawn of host bacterial cells are spotted with a small volume of 5–10 µL enzyme suspension. The minimum amount of endolysin which shows clear bacterial cell lysis is defined as the “minimum inhibitory concentration” of that endolysin [30]. Until now, most of the characterized endolysins showed good bactericidal activity ranging from 10^2^–10^8^ units/mg of the enzyme [51].

## 4. Regulation of Expression

Bacteriophages are dependent on their bacterial hosts, as the lytic and lysogenic life cycles are intervened with at the genetic level by utilizing host genetic machinery [1,52,53]. Considering the importance of the bacterial host in the regulation of phage enzyme expression, it is necessary for phages to hijack host genetic machinery for their progeny generation, as well as their liberation through host cell by the production of endolysins. Hence, it is necessary to understand how the phage genome regulates the expression of different phage enzymes during their lytic life cycle [1,52]. Endolysins and holins are an important group of phage lytic proteins that mediate the two critical processes during phage life cycle, namely “lysis from within” (phage releasing from host cell) and “lysis from without” (endolysin present on some phage surface lyse host cell) [54,55]. The regulation of phage lytic proteins like holins, VAPGHs, and endolysins (lytic genes transcription) occurs during the different stages of phage life cycle and is maintained until the end. The lytic enzymes are produced to release the phage progeny from the host cell, which is reported from the different experiments carried out with different phages [56,57,58,59,60]. The transcriptions of endolysin are carried out by host transcription factors [61,62] and post-translational regulation of endolysins is also reported, where holin-mediated transport or mechanisms not yet fully understood are involved in endolysin transportation in the inactive form, which later becomes active after protein conformational changes [52].

The regulation of expression of phage lytic enzymes in Lambda phage (λ phage) explains the mechanism by which its lytic and lysogenic life cycle are controlled by Cro and cI proteins [63,64]. For the phages to undergo the lytic cycle, the transcription of Cro gene to produce Cro protein is essential. For the Cro protein to express, the absence of cI protein is crucial. Whereas, for the lysogenic cycle, the transcription of cI gene to produce cI protein is essential, and for this protein to express, the absence of Cro protein is decisive [63,64]. The nutritional condition of the host cell also determines the lytic or lysogenic decision [53]. In the presence of nutrients, a lytic state is established and phage progenies are released, whereas the absence of nutrients leads to a lysogenic conversion [53]. The initial regulation of Cro protein is essential for the expression of different lytic genes, which also involves phage lytic proteins such as endolysins, holins, and virion-associated peptidoglycan hydrolases (VAPGHs) [65]

However, in lytic phages, expression of endolysins and lytic proteins are independent of Cro gene making them effective for phage therapy [63,64,65,66]. On the contrary, it is interesting to know that, unlike lytic phages, lysogenic phages integrate their genetic material in the host genome and multiply with the host genome until they undergo lytic conversion [56,57,58,59,60]. This suggests that the evolution of phage lytic enzymes and proteins is more frequent in lysogenic phages rather than lytic phages, as the latter lack the essential enzymes required for horizontal gene transfer and end up degrading the infected host genome following lytic activity [63,64,65,66].

## 5. Endolysins as Antibacterial Agent

Since the discovery of bacterial viruses, known as bacteriophages, their lytic enzymes (endolysins) have also received significant attention, particularly for their potential as an alternative therapeutic against bacterial infections in animals and humans [67].

### 5.1. Human Medicine

Due to the reduction in antibiotics efficacy, many infectious pathogens have become life-threatening agents in humans. Therefore, to combat this issue, research has focused on phage-derived endolysins for both topical and systemic infections in humans [68].

The human skin is a natural barrier to prevent microbial infections; however, disruption of the epithelial barrier (e.g., cuts, burns, wounds, etc.) may result in infections caused by both Gram-negative and Gram-positive species [69,70]. One particular Gram-positive pathogen that has received significant focus is *S. aureus*, due to its involvement in topical skin or tissue infections, as well as systemic blood poisoning, bone, and cardiac infections [71]. Furthermore, the rise in multidrug-resistant (MDR) and methicillin-resistant *S. aureus* (MRSA) has reduced the availability of effective therapeutics. Therefore, recombinant endolysins have become a novel treatment, paramount to controlling *S. aureus* super-bugs in clinical or hospital sectors. For example, commercially available recombinant endolysin, Staphefekt SA.100, and XDR.300 (Micreos Human Health BV, The Netherlands) have been implemented in patients with chronic skin disease caused by *S. aureus* [72]. The administration of Staphefekt has been advised in the presence of signs and symptoms of inflammation for most of the skin infections including rosacea, acne, and eczema [73]. Another endolysin, CHAP_K,_ has the potential to reduce *S. aureus* colonization in the skin; therefore, this novel endolysin is used as a disinfecting agent in healthcare sectors [74].

*S. aureus* has also been considered as a major problem in nasal infections. *S. aureus* is the causative agent in up to 30% of human nasal infections worldwide [75]. Phage endolysin P128 has been formulated into a hydrogel and in vitro assays show this formulation can successfully lyse clinical *Staphylococcal* isolates collected from the human nostrils [76]. However, unfortunately, endolysins have not yet been permitted for use in reducing *S. aureus* contamination in the human nasal cavity and are awaiting clinical trials [77].

In contrast, Gram-negative pathogens such as *Acinetobacter baumannii* and *Pseudomonas aeruginosa* have been considered major opportunistic pathogens in burn wounds [78]. To combat such drug-resistant pathogens, novel engineered endolysin named artilysins have been suggested as suitable alternative agents (Artilysin^®^ is a registered trademark in the European Union, United States, and other countries). In 2014, Briers and their research team have successfully evaluated novel endolysin LoGT-008 activity with minimum inhibitory concentration against *P. aeruginosa* and *A. baumannii* responsible for skin infections [79]. The developed human neonatal keratin epidermal cell line model was challenged with *P. aeruginosa,* followed by administration of LoGT-008 and a wild type endolysin PVP-SE1gp146. Results showed that both endolysin components could protect the human cell line (100%) from *P. aeruginosa* infections via the rescue of the cell line and reduced bacterial numbers [79].

Phage-derived antimicrobial compounds have been successfully applied against most of the Gram-positive pathogenic infections. Recent studies have revealed the efficiency of phage lysins against *S. aureus* pathogen using animal experiments and human clinical cases [80]. Interestingly, phage endolysins are applied in various human clinical trials right now. For instance, endolysin SAL200 and CF-301 have been successfully developed against *S. aureus* for the treatment of bloodstream and cardiac infections [67]. A decade ago, Cheng and Fischetti addressed the phage muralytic enzyme, PlyGBS, with weak lytic activity compared to its mutant PlyGBS90-1 with hyperthymic activity. Research proved that the population of streptococci could be reduced by considerable numbers (28-fold) using a phage muralytic enzyme mutant. Bacterial vaginosis is the best example of the phage-derived protein approach. Moreover, this research team noticed that the mutant PlyGBS90-1 has numerous benefits such as rabid, efficient lytic activity against streptococci in mouse vaginal colonization model [49]. This new system with phage-derived lysins would be a robust approach to declining antimicrobial resistance compared with phage cocktail therapy (Figure 5).

### 5.2. Veterinary Sector

Reports state that penicillins are the most prescribed antibiotics in the veterinary sector [81,82]. Due to the misuse and overuse of antibiotics, we have seen the dramatic evolution of bacterial resistance in aquaculture, agriculture, and veterinary medicine [83,84]. The transmission of antibiotic-resistance genes from contaminated food to humans and vice versa has been extensively studied [85]. Particularly, food animals such as cattle, poultry, and swine have been found to be a major reservoir of antibiotic-resistant bacteria and its specific gene that can move to people directly or indirectly by the food chain [86]. As a consequence, the use of antibiotics in animal feed has been banned in the United States [84] and the European countries [87]. Therefore, there is a significant need for antibiotic alternatives, such as endolysins, along with the preclinical studies of endolysins for veterinary applications [88].

Phage-encoded endolysins have been recommended as an impressive agent to counteract most farm animal-related pathogens such as *Clostridium perfringens*, *Streptococcus suis*, *Paenibacillus larvae,* and *Salmonella* species [88]. The poultry industry has reported major issues due to the emergence of Gram-positive antimicrobial-resistant pathogen *C. perfringens,* which can cause infections in up to 95% of chickens [89]. In 2014, Tamai and their research team have also performed and designed phage endolysin phiSM101, that has been utilized in poultry with broad lytic activity against *C. perfringens* pathogen [90]. In that same year, Gervasi and colleagues designed an amidase endolysin, CP25L, which harbored the ability to reduce 50% of another Gram-positive pathogen *Lactobacillus johnsonii* within 24 h [91]. Similarly, salmonellosis is the major infection in poultry caused by Gram-negative pathogen *Salmonella* that is also relatively common and results in economic losses in the poultry industry [92]. In general, research proves that phage endolysin has the ability to defeat antibiotic resistance and its problems in both Gram-positive and Gram-negative pathogens.

A deadly zoonotic disease, anthrax has the potential to infect farm animals and wild animals as well as humans. Schuch and colleagues reported the advantages of amidase-type endolysin PlyG isolated from gamma phage-type with significant therapeutic effect for the control of *Bacillus anthracis* [93]. Despite such promising results, a few major challenges should be considered for the application of endolysins in the veterinary area, particularly cost and production efficacy. In recent studies, such challenges of endolysin have been improved by the modern protein modification assay and algae-based endolysin synthesis assay, respectively [94].

*Streptococcus equi* is the primary representative for upper respiratory tract inflammation in *Equus caballus* (horses) [68]. To control *S. equi* pathogen, Hoopes and colleagues have explored the potential of amidase endolysin PlyC, which can broadly be used as an antimicrobial agent with 1000-fold increase in efficacy compared to routinely used disinfectants [95].

Canine pyoderma is one of the most common bacterial skin infections diagnosed in dogs, which is generally caused by methicillin-resistant *Staphylococcus pseudintermedius* (MRSP). An earlier clinical trial showed that dogs can recover from pyoderma with continuous treatment with anti-staphylococcal protein P128 hydrogel [96]. In addition, chimeric endolysins have been used with the combination of the cell wall-targeting domain of bacteriocin and the muralytic domain of phage K to treat canis skin lesions [76].

Bovine mastitis is the most prominent and persistent disease in animal husbandry, and three-quarters of bacterial infections are caused by *Staphylococci* and *Streptococci* species [97]. In previous studies, Schmelcher and colleagues demonstrated the synergistic effect of phage lysins with lysostaphin to combat *S. aureus* in the murine animal model [30]. Over recent years, pure endolysin Trx-SA1 of phage IME-SA1 has been used in the therapeutic trial process to control clinical mastitis-caused MRSA [98]. Another notable zoonotic microbe, *Streptococcus suis*, has been associated with pig-related diseases such as arthritis, septicemia, meningitis, and endocarditis [68]. Surprisingly, Wang and colleagues have proved that *Streptococcus* phage endolysin, LySMP, can recover diseased pigs from clinical *S. suis* in China [99]. While the use of endolysins as an animal therapeutic is currently in its infancy, such promising results pave the way for future research within this field.

### 5.3. Agriculture and Plants

The prevalence of antibiotic resistance in the food chain process within agriculture and crop culture has led to the causes of bacterial infection in humans. For example, multidrug-resistant leaf blight rice can cause nosocomial infections in humans with a weak immune system. Thus, phage-based endolysin therapy has been suggested to ensure the safety of plants from pathogens. Two decades ago, transgenic tomato plants were successfully produced with CMP1 phage endolysins to prevent infection of *Clavibacter michiganensis,* a bacterial species responsible for bacterial canker [100]. Similarly, another team focused on transgenic potato plants produced with T4 phage lysin that holds resistance against the pathogen *Erwinia carotovora* [101]. A previous report revealed that the endolysin-based defense system has significant potential to overcome antibiotic resistance with the design of transgenic plants.

*Apis* (Honey bees) are significant insect pollinators of crops; however, they are commonly infected with *Paenibacillus larvae*, which causes sepsis and death [68]. To control the emergence of resistance, Endolysin PlyV12 has potential with high lytic activity against antibiotic-resistant *Enterococcus faecium* and *Enterococcus faecalis* [102].

### 5.4. Food and Biotechnology

#### 5.4.1. Food

It is well known that food animals such as chicken, cattle, and pigs and their products are a source of drug-resistant pathogens [68]. To control the emergence of resistance, Endolysin PlyV12 has potential with high lytic activity against antibiotic-resistant *Enterococcus faecium* and *Enterococcus faecalis* [102]. Interestingly, plant-based milk is susceptible to *L. monocytogenes* contamination; therefore, studies have shown excellent sterilization efficiency in soya milk when LysZ5 is administered [103]. Additionally, *L. monocytogenes* pathogen has successfully been treated with various alternate phage endolysins such as PlyP825, PlyP40, and Ply511 in the presence of hydrostatic pressure [104].

Throughout this review, *S. aureus* has been highlighted as a pathogen in human and animal medicine; however, this bacterium is also responsible for food and milk contamination during the production process [68]. In a recent study, Chang et al. reported that the existence of a cell wall binding domain in *Staphylococcal* endolysin, LysSA11, showed excellent specificity and antimicrobial activity when compared to endolysin LysSA97 with moderate activity against *S. aureus* [105]. Overall, phage endolysins have been raising interest due to their various applications in food safety and food processing systems.

#### 5.4.2. Biotechnology

Advanced biotechnological approaches such as genetic engineering and therapeutic protein (endolysin) have the potential to improve the food safety process via rapidly eradicating bacterial pathogens due to their enzymatic nature. To stop the prevalence of antibiotic-resistant microbes that cause food-borne infections, phage endolysins have been recognized as natural bio-preservatives in food sectors. During food production, it is necessary to ensure safe and hygienic surfaces are maintained, in order to prevent foodborne outbreaks within the community. Using peptidoglycan hydrolases and endolysins, biofilms have been completely eradicated in food and clinical environments, ensuring sufficient surface disinfection.

Bacterial biofilms’ associated problems can be overcome by engineered endolysin or peptidoglycan hydrolases. For instance, engineered endolysin SAP-2 and LysSMP have the potential to eliminate biofilms formed by *Staphylococcus* and *Streptococcus* species, two important pathogens in food and clinical environments [106,107]. Biofilm matrix can be directly lysed by endolysin; that happens due to its diffusion through the extracellular material of pathogens. Moreover, endolysins have significantly been involved in developing vaccines. In this process, a phage endolysins-based surface display system has been applied to lactic acid bacteria, specifically in food and medical applications [108].

Recombinant endolysin has been applied for the treatment of farm animals. For instance, MRSA cell numbers have been effectively reduced when endolysin Trx-SA1 was used as a therapeutic tool in cow udders. In addition, animal model study proved that the combination of phage endolysins (2638A, 80a, LysK, lysostaphin, phi11, and WMY) can provide complete protection from bacteremia-induced death when compared with single endolysin (phiSH2) therapy. It is clear that phage endolysin with biotechnological approach can assure the control of future farm animal infections [65].

In the case of clinical trials, advanced techniques like extensive protein engineering have enhanced phage endolysin activity against antibiotic-resistant Gram-positive and Gram-negative pathogens. Furthermore, unique endolysin domain has been applied for various purposes, for instance, immune-based therapeutic and diagnostic approaches [69]. Phage endolysin-based biotech companies such as GangaGen, ContraFect, and Micreos have developed phage lytic proteins P128, CF-301, and Staphefekt against *S. aureus*, respectively. Those endolysin products can successfully treat chronic kidney disease, bloodstream infections, and skin infections [19,65]. Overall, phage endolysins have been considered to be a promising approach to diagnosing and killing pathogenic populations in the most crucial fields such as agriculture, human medicine, and veterinary medicine today (Figure 6).

## 6. Endolysin in Biofilms Eradication

Bacteria are universally found in nature attached to surfaces such as living tissues, medical devices, industrial equipment, or food [109]. During attachment, some bacteria produce extracellular polymeric substances (EPS) forming a complex cluster of bacterial cells, known as biofilm [109]. This polymeric network mainly consists of exopolysaccharides as well as nucleic acids, proteins, and lipids, providing mechanical stability and adhesion to surfaces [110]. The life cycle of the bacterial biofilm is shown in Figure 7.

In clinical and food settings, biofilms are major concerns as they form on critical locations causing contamination that affects the efficacy of the established procedures; for example, the bacterial colonization on the outer surfaces of catheters [111]. Moreover, they cause treatment failure in surgeries and chronic wounds due to antibiotic-resistant bacteria housed within the biofilm network [112].

EPS matrix reacts chemically with the antimicrobial agents and limits their diffusion rate [113]. In addition to the EPS matrix protection, the biofilm cells change the surrounding environment to provide conditions that inactivate any antimicrobial agent and protect the microbial communities within the biofilm network. Consequently, the development of novel anti-biofilm techniques has been necessarily required to provide further control strategies [111].

Lytic phages are used in phage-based therapies not only because they damage the bacterial hosts but also because they lack the essential enzymes for horizontal gene transfer [66]. Many characteristics should be taken into consideration during designing phage-based methods to treat biofilms and bacterial infections such as the rate of phage diffusion, penetration, and propagation [114]. Endolysins are very useful in biofilms treatment; their use has shown significant results and their efficacy has been discussed in many studies. Guo, M. et al. used the novel endolysin LysPA26 to eliminate *Pseudomonas aeruginosa* in the biofilm formation [66]. LysPA26 could lyse other Gram-negative bacteria such as *Acinetobacter baumannii*, *Klebsiella pneumonia,* and *Escherichia coli* under a broad range of temperatures from 37 °C to 50 °C. Their study demonstrated that LysPA26 could degrade the biofilm and disrupt the bacteria in a concentration-dependent manner; this was indicated by a reduction in the biofilm optical density (OD_600_) [66]. Meng et al. experimented with the effect of manufactured bacteriophage lysin, LySMP, to treat *Streptococcus suis* biofilm alone and mixed with antibiotics and bacteriophage. They found that LySMP alone could treat the biofilm with >80% removal, compared to <20% removal when the biofilm was treated with bacteriophage alone and/or with antibiotics. The results demonstrated that LySMP could act synergistically to treat *S. suis* biofilm in a concentration-independent manner and inactivate the released cells [106].

In animal models, the engineered peptidoglycan hydrolase is used to cleave important bonds in the peptidoglycan structure of *S. aureus* that increased the rate of bacterial colony lysis and biofilm removal [115]. Recently, glycoside hydrolases show results to disturb *P. aeruginosa* biofilms and encourage the killing of neutrophil-mediated communities [115]. In addition, fusion proteins that are derived from bacteriophage-encoded endolysins can specifically reduce the resistance rate of bacterial communities in the biofilm without targeting the commensal bacteria [116].

Persister cells are small subpopulations of bacterial cells that show high resistance to antibiotics because of their ability to enter a dormant state when they are treated with a bactericidal antibiotic that enables them to survive [117]. Therefore, persister cells are one of the big issues in biofilm removal. In addition to their antibiotic resistance activity, they can regenerate the population just after stress removal [118]. Gutiérrez et al. (2014) used phage-derived lysin, LysH5, in biofilm removal to kill *S. aureus* persister cells; results showed no persister cells remained in the *Staphylococcal* biofilm treated with 0.15 μM of LysH5. There was also a complete inhibition in biofilm formation in select strains.

## 7. Immunogenicity, Safety, and Resistance

In the wake of the urgent need to develop alternatives to conventional antibiotics for therapeutic use, bacteriophages have been considered as plausible alternatives. However, their use as antimicrobial agents might be limited or totally hindered due to limitations they pose which include: reduced activity due to immune system response, the possible emergence of bacterial resistance against bacteriophages, and health safety issues [119]. While the myriad of limitations involved in the use of whole phage cause concerns, attention is fast shifting to the use of phage endolysins. As promising as endolysins are in the fight against antimicrobial resistance, their ultimate approval as therapeutics for the public is highly hinged on these three factors; immunogenicity, which is how the immune system of the body will respond to them, their safety profile on human health, and the possibility of bacteria building resistance against endolysins. Such factors require clinical trials on endolysins for a better insight into these concerns.

These factors and concerns are highly justifiable as most biopharmaceuticals are known to induce immune responses; in some cases, the consequences can be severe and potentially lethal, causing a loss of efficacy of the drug or even worse, leading to autoimmunity and hypersensitivity reactions. The body elicits an immune response against exogenous protein products, a class to which phages and endolysins belong, by the activation of T cells and the consequent production of neutralizing antibodies [120]. In fact, this effect has been one of the limiting factors in the use of phages therapeutically [121,122]. Antibodies are known to be poorly effective in endolysin inactivation; an example is seen in a study carried out using Cpl-1 where the presence of antibodies sufficiently reduced the systemic half-life of Cpl-1 to approximately 20 min [17]. The result of a study carried out using LysGH15 endolysin to test for the effect of endolysin inactivation by antibodies shows that LysGH15-specific antibodies did not affect the killing efficiency of LysGH15 against MRSA in vitro or *in-vivo* [123]. Similar studies conducted by other researchers have shown that this trait is not specific to LysGH15. In a study conducted by Loeffler et al. [17], using endolysin Cpl-1 on pneumococcal bacteremia, rabbit hyper-immune serum raised against Cpl-1 was used to determine the effect of immunized serums on endolysin-bacterial lysis. The result of the research showed that in non-immunized serum, 2000 µg of Cpl-1 decreased the viable bacterial count in a pneumococcal solution from log10^9.2^ to log10^6.2^ CFU/mL within 1 min. However, when the same experiment was performed using hyper-immunized serum (titer = 10,000), the bacterial titer decreased slightly less, to log10^7.0^ CFU/mL. All titers decreased another 0.5 log10 CFU/mL by 10 min; this indicated that high immune serum decreased, but did not completely block the killing of *S. pneumoniae* by Cpl-1. Although antibodies are known to be poorly effective in endolysin inactivation, their presence sufficiently reduced the systemic half-life of Cpl-1 to approximately 20 min, indicating that one or two doses of endolysin therapy might not be enough for complete eradication of infections [17]. Concerning safety issues, the pre-clinical trials performed using animal models have shown that endolysins have a good safety profile and showed no adverse effects like fever, abdominal pain, or diarrhea, proving to be safe moving forward [67].

To support the safety of endolysins, SAL200 was administered intravenously to dogs and monkeys, and no adverse effects were observed or reported in both animals [67,124,125]. The safety profile and tolerance of SAL200 endolysins were also evaluated in humans; adverse effects reported in more than three participants were fatigue, rigors, headache, and myalgia. Furthermore, no clinically significant values were recorded with respect to the findings of clinical chemistry, hematology and coagulation analyses, urinalysis, vital signs, ECG, and physical examinations [67].

Another study conducted to test the effect of LysGH15 on *S. aureus* in mice showed that high-dose LysGH15 injection did not cause significant adverse effects or pathological changes in the main organs of treated animals [123]. LysGH15 does not enhance IgE levels among total serum antibodies. IgE activates mast cells and basophils, and these cells release histamines and other chemical substances upon activation, an essential step during most allergic reactions. Histological analysis of the study showed that neither repeated injections nor large-dose infusions of LysGH15 resulted in inflammation or mast cell activation in major organs in mice. The potential for an IgE response in humans should be investigated, as this effect would trigger severe side effects such as hypersensitivity, allergic reactions, and possibly cancers if extreme [123].

A study conducted by Jun et al. tested the effect of SAL200 endolysin on *Staphylococcal* infections in humans. Results showed that the potential of developing resistance to SAL200 is significantly lower than that of conventional antibiotics [67]. The report further showed that no mutants were resistant to SAL200 during its development. Repeated exposure of an *S. aureus* strain to half of the minimum inhibitory concentration (MIC) of SAL200 30 times failed to generate a resistant mutant [67].

This is indeed great news, as in response to phage, bacterial cells can evolve mechanisms of resistance [126]. Such resistance tactics include modification of phage receptors on the bacterial surface, secretion of substances that prevent phage adhesion to the bacterial pathogen, activation of measures for blocking phage DNA injection into the cell, and inhibition of phage replication and release [126]. In fact, statistically, in all bacterial populations, such resistant mutants exist, and they can become prevalent due to the selective pressure by bacteriophages during phage therapy [127].

## 8. The Effect of Endolysins on the Normal Microbiota

The human host is colonized with a large number of microbial cells as part of their normal microbiota, covering different parts of the human body such as the urogenital and gastrointestinal tracts, the nasal and oral cavities, and the skin surface [128]. It is estimated that more than 100 trillion symbiotic microorganisms colonize human beings and are of great significance to human health and illness [129]. The normal microbiota forms a physical barrier to protect its host from foreign pathogens, through competitive exclusion and antimicrobial production [130]. Any disturbance to the normal microbiota may cause serious disease. For example, intestinal microbiota dysbiosis influences the level of immune mediators’ production leading to metabolic dysfunction and chronic inflammation [131]. Additionally, hepatitis B virus (HBV), human immunodeficiency virus (HIV), and other diseases are associated with microbiota disturbance [131].

Even though antibiotics act to improve human health and life expectancy, broad-spectrum antibiotics disrupt the existing microbiota, causing dysbiosis and leading to disease outcomes [128]. Figure 8 shows the effect of microbiota disturbance on different systems. In contrast to antibiotics, endolysins selectively treat specific species or subspecies of pathogenic bacteria without causing disturbance to the surrounding normal microbiota [102], as previously reported by experiments on *Enterococcus faecalis* and *E. faecium* using bacteriophage-induced lysin called PlyV12. Their results demonstrated that PlyV12 showed a great lytic effect on the vancomycin-resistant strains of *E. faecalis* and on multiple strains of *E. faecium*. Therefore, lysins show less disruption of the normal microbiota when they are used to treat various infections, as they do not transfer resistance genes or bacterial toxins destroying the colonizing bacteria of mucous membranes [132]. They can induce the response of the immune system without neutralizing or preventing antimicrobial activity. Hence, they can be used for systemic infections treatments [133]. Additionally, lysins treat the rapid lysis of Gram-positive bacteria through an exogenous application as recombinant enzymes [132].

For instance, an enterococcal lysin is reported to kill enterococci and a number of other Gram-positive bacteria, including *S. aureus*, *S. pyogenes*, and group B streptococci [102]. Despite their low activity toward these pathogens, this enterococcal lysin has been recorded as one of the broadest acting lysins recognized [102]. Additionally, some naturally occurring lysins have potent activity toward Gram-negative bacteria despite the structural barrier and low permeability of their outer cell membrane to lysins. Ghose and Euler (2020) stated that the addition of some stimulants increases the intrinsic bactericidal activity of lysins significantly toward Gram-negative bacteria. For instance, the fusion protein called artilysin that is composed of a lysine fused to destabilizing peptide is active against both Gram-positive and Gram-negative bacteria. One example of artilysins is the broad-spectrum Art-175 that is composed of *P. aeruginosa* lysin KZ144 modified version [79,134,135]. Therefore, using lysins in infection control is recently of great importance for treating different specific infectious diseases without causing serious harm to the normal microbiota [133]. Imanishi and colleagues conducted experiments **in-vivo** and in vitro to examine the lethal effect of the Kayvirus-derived endolysin on *staphylococcal impetigo* [136]. The results showed that the enzyme could exhibit a bactericidal effect on *S. aureus* after 15 min incubation period and reduced intra-epidermal staphylococci number and pustules size in the impetigo mouse. Additionally, the treatment with lysin increased the skin microbiota diversity in the same animal model.

## 9. Endolysins and Antibiotics

The increased difficulty in treating antibiotic-resistant bacterial infections has created the need for novel antimicrobial treatment options, which can be used as stand-alone therapeutics or as a complement to antibiotics [137]. Research endeavors over the years have raised strong recommendations for the synergetic approach to bacterial treatment via endolysin–antibiotics combination. Early research work by Beveridge and colleagues describe the role of Gram-negative bacteria outer membrane in shielding bacterial peptidoglycan cell wall from exogenous treatment [138]. This work paved the way for subsequent research works, which devised methods to breach the outer membrane protective mechanism [138]. In ensuring that the bacterial outer membrane barrier was no hindrance to the exogenous lytic action of endolysin, Thumeepak et al. deployed the combined effect of endolysin and antibiotic therapy in the treatment of multi-drug resistant strains of Gram-negative *Acinetobacter baumanii* [139]. The study revealed elevated levels of antibacterial activity with the combined administration of endolysin LysABP-01 and the antibiotic colistin, compared to either of the treatments alone [139]. Colistin, also known as Polymyxin E, contains positively charged molecules that interact with negatively charged phosphates of the cell membrane and displace divalent cations (Ca^2+^ and Mg^2+^), which contribute to outer membrane instability. This interaction results in damage to the outer membrane and leakage of cellular components causing eventual cell death [140]. The membrane destabilizing activity of the antibiotics is then exploited by LysABP-01 to promote access towards the cell wall target. The combination therapy was very successful, as there was a significant increase in antibacterial activity, with growth inhibition rate reaching up to 100%. There was a considerable reduction in the minimum inhibitory concentrations (MIC) of LysABP-01 by 32-fold and that of colistin by 8-fold, which allowed for lesser doses of the two agents to be administered for treatment.

This approach has been proven to be equally effective in the treatment of Gram-positive strains of Vancomycin-intermediate *Staphylococcus aureus* (VISA), through the combined use of anti-Staphylococcal endolysin MV-L (derived from novel *Staphylococcus aureus* bacteriophage ØMR11) in combination with glycopeptide antibiotics such as vancomycin or teicoplanin [141]. Individually, MV-L at 50 U concentration and vancomycin at 4 μg/mL concentration showed weak suppression of Mu50 VISA strain growth rate. In contrast, the simultaneous administration of the agents (12.5–50 U of MV-L and 4 μg/mL of vancomycin) lysed the Mu50 cells to a greater extent. This synergy was also confirmed in broth culture, with a predicted 100-fold decrease in colony-forming units per millimeter for the combined treatments [141]. Authors attributed the synergy observed to the partial degradation of the thickened bacterial cell wall by the MV-L endolysin, which allowed glycopeptide access to sites close to the cell membrane.

In a similar study conducted by Kim et al. the anti-*Staphylococcal* effect of phage endolysin SAL200 combined with standard-of-care (SOC) antibiotics such as nafcillin and vancomycin was examined [142]. The study assessed in vitro bactericidal activity and subsequent *in-vivo* activity using murine and *Galleria mellonella* infection models. There was a significant decrease in the antibiotics’ minimum inhibitory concentration (MIC) and *S. aureus* concentration in both assays. The study revealed the effectiveness of SAL200 in restoring the sensitivity of nafcillin and vancomycin antibiotics at the brink of resistance [142]. A lower bacterial density was reported in the splenic and blood tissues in the murine model treated with a combination of SAL200 and SOC antibiotics, compared with SAL200 treatment only. *Galleria mellonella* larvae infected with methicillin-resistant and methicillin-susceptible strains of *Staphylococcus aureus* were treated with SAL200 and SOC antibiotics at 96-h post-infection time, which resulted in an increased survival rate for both strains. Overall, such results validate the synergetic relationship between SAL200 and SOC antibiotics, both in vitro and *in-vivo*.

Letrado et al. demonstrated the synergetic effect of Cpl-711 endolysin and commonly used antibiotics such as amoxicillin, levofloxacin, vancomycin, and cefotaxime in the treatment of multi-drug resistant strains of *Streptococcus pneumoniae* [143]. The resulting minimum inhibitory concentration indices revealed a robust synergetic relationship when endolysin Cpl-711 was combined with amoxicillin and cefotaxime. The group suggested that a plausible explanation for the synergism observed was the breakdown of the peptidoglycan cell wall by antibiotics, which resulted in increased susceptibility to endolysins; a mechanism quite similar to that earlier described by Rashel et al. [141].

A study conducted by Becker et al. highlighted the synergetic role of endolysin Lys K and lysostaphin, a bacteriocin secreted by *Staphylococcus simulans* in killing *Staphylococcus aureus* [144]. Both antibacterial agents, which had been initially proven as effective in the treatment of multi-drug resistant *S. aureus* strains in two separate studies were successfully combined using a checkboard assay [145,146]. The pattern of cleared wells indicating growth inhibition confirmed greater antimicrobial strength in combined use. To support this, individually, each agent had MIC values of <30%; 18% for lysostaphin and <16% and 33% for Lys K endolysin for *S. simulans* and *S. aureus*, respectively. The overall fractional inhibitory concentration (FIC) of 0.449 ± 0.069 on the other hand, however, suggests a robust synergetic relationship [144].

In a more recent study conducted by Kashani et al., cysteine/histidine-dependent amidohydrolase/peptidase (CHAP) and amidase, which are catalytic domains of endolysin Lys K, were synergistically combined with vancomycin in the treatment of methicillin-resistant *Staphylococcus aureus* (MRSA). The calculated FIC index (∑) revealed an 8-fold reduction in the MIC value of vancomycin due to the synergistic effect of both agents using a two-fold dilution [137].

Overall, the lytic action of antibiotics on bacterial cell wall peptidoglycan and the consequent increase in endolysin susceptibility is believed by many to be the basis of endolysin–antibiotics synergy, and this has been experimentally substantiated in studies detailed in this review [141,143]. The evidence presented in these studies affirms the role of endolysin–antibiotic synergy in improving therapeutic efficacy, as shown in a marked decrease in antibiotic minimum inhibitory concentration. This, of course, would translate to a reduction in antibiotics dosage administration. The remarkable results of the above studies present high hopes in antibacterial treatment, as it becomes progressively evident that the war against antimicrobial resistance requires a more inclusive approach, combining novel knowledge in endolysin research with conventional antibiotic use. The empirical pieces of evidence from these studies are undeniable; they, however, would require human clinical trials before they can be fully adopted as standard care practice.

## 10. Formulation and Administration Routes

To perform a successful administration of any therapeutic agent to the target site of infection, there are some issues to be considered, including the administration route and suitable delivery systems that maintain the treatment’s stability and activity [70]. Currently, the administration of phage-derived enzymes can be applied via different routes including injections (intravenous, and intraperitoneal), topical applications (creams, ointments, and gels), transnasal, vaginal, and oral delivery systems [147]. Table 1 summarizes the different endolysin administration routes and the targeted pathogenic bacteria.

The oral route of phage endolysins has the challenge to preserve the enzymatic activity due to harsh gastric proteolytic enzymes and acidic pH. It is intended that the encapsulation of phage proteins could provide a protective strategy to solve this problem [148]. Based on an initial human phase study of SAL200, the endolysin has proven to be safe and effective besides causing mild adverse effects such as fatigue and headaches [21]. Information was obtained on the pharmacokinetics and pharmacodynamics of the product upon intravenous injection in human subjects [67]. SAL200 is the first endolysin-based therapeutic formulation with a recombinant form of phage endolysin SAL-1 (rSAL-1) [21]. This phage endolysin has in turn been derived from the bacteriophage SAP-1, which infects several Staphylococci strains such as MRSA and vancomycin-resistant *S. aureus* (VRSA) [18,149].

**Table 1 antibiotics-10-00124-t001:** Selection of some endolysins and their administration routes.

Target Pathogen	Phage	Enzyme (Endolysin)	Activity (Mode of Action)	Administration Route	References
***Streptococcus pneumoniae***	Cp1	Cpl-1	Muramidase	Intravenous, nasal, oral, aerosols, and Intraperitoneal	[17,150,151,152,153,154]
	Dp-1	Pal	Amidase	Nasal and oral	[50,155]
	CP-7	Cpl-7	Muramidase	Immersion	[156,157]
***Streptococcus pyogenes***	C1	PlyC	Amidase	Oral, nasal	[16]
	MGAS5005 prophage	PlyPy	Endopeptidase	Intraperitoneal	[158]
***Streptococcus agalactiae***	NCTC11261	PlyGBS	Endopeptidase and Muramidase	Intravaginal, oral and intranasal	[159]
	SK1249 prophage	PlySK1249	Amidase and endopeptidase	Intraperitoneal	[160]
**MRSA**	GH15	LysGH15	Amidase and endopeptidase	Intravenous and Intraperitoneal	[123,161]
	MR11	MV-L	Amidase and endopeptidase	Intraperitoneal, nasal	[141]
	K	LysK	Amidase and endopeptidase	Intraperitoneal	[162]
	SAP-1	SAL-1	Amidase and endopeptidase	Intravenous	[18]
	phiSH2 prophage	phiSH2	Amidase and endopeptidase	Intraperitoneal	[162]
	phiP68	P68	Amidase and endopeptidase	Intraperitoneal	[162]
	phiWMY	LysWMY	Amidase and endopeptidase	Intraperitoneal	[162]
	phi80α	80αLyt2	Amidase and endopeptidase	Intraperitoneal	[162]
	phi11	phi11	Amidase and endopeptidase	Intraperitoneal	[162]
	2854 prophage	2638A	Amidase and endopeptidase	Intraperitoneal	[162]
***Pseudomonas aeruginosa***	phage PVP-SE1	Artilysin^®^ Engineered Endolysin-Based (PVP-SE1gp146)	Muramidase	Oral and topical	[79]
	*P. aeruginosa* phage	PlyPa03	Muramidase	Topical	[163]
	*P. aeruginosa* phage	PlyPa91	Muramidase	Intranasal	[163]
***Acinetobacter baumannii***	RL-2015	PlyF307	Muramidase	Intraperitoneal and Topical	[164]
	SS3e	LysSS	Muramidase	Intraperitoneal	[165]
***Bacillus anthracis***	γ-phage	PlyG	Amidase	Intraperitoneal	[93]
*** Enterococcus faecalis ***	*E. faecalis* phage IME-EF1	LysIME-EF1	Endopeptidase	Intraperitoneal	[166]
	*E. faecalis* phage EF-P10	LysEF-P10	Endopeptidase	Intraperitoneal	[167]

## 11. Insights into the Clinical Trials of Engineered Endolysins

Recent advances in the sequencing of phage genomes have ignited the application of endolysins as antimicrobials [45,168]. Endolysins, when applied to Gram-positive bacteria, cause immediate results. In Gram-negative bacteria, the peptidoglycan layer (susceptible layer) is protected by a less permeable outer membrane. Various strategies have been adopted to address this issue, including the use of outer membrane permeabilizers such as EDTA, gallic acid, and thymol in combination with endolysins in Gram-negative bacteria to provide significant results [169]. permeablilizer

Endolysins combined with fusion peptides (lipopolysaccharide destabilizing peptides) also promote the uptake of endolysin through the outer membrane in Gram-negative bacteria. Artilysin^®^ Art-175 (LYSANDO AG, Liechtenstein) is very effective against multidrug-resistant strains of *Pseudomonas aeruginosa* and *Acinetobacter baumannii* [170]. It is made by fusing the antimicrobial sheep myeloid peptide (29 amino acids) with endolysin KZ144 [79].

Modification of endolysins by molecular engineering techniques also improves the lytic activity of an enzyme. Endolysins can have two types of domain structure i.e., globular (Gram-negative bacteria) and modular (mainly Gram-positive bacteria). The modular structure comprises N-terminal enzymatically active domain (EAD) and C-terminal cell wall-binding domain (CBD) linked together by short, flexible regions. N-terminal serves the purpose of enzymatic hydrolysis, and C-terminal is for substrate recognition [68]. Studies have shown that deletion or shortening of C-terminal CBD results in enhanced lytic activity [171].

Both phages as well as isolated and bioengineered endolysins assure the therapeutic potential against pathogenic bacteria. Endolysins have a broader host range than phages [172], and still, due to their specificity, they only target specific pathogenic bacteria without causing damage to microflora. Until now, no case of endolysin resistance has been reported. This is because their target layer (peptidoglycan layer) is highly important for the viability of bacteria and any mutation can lead to serious damage to bacteria [65].

Preclinical and clinical trials for therapeutic endolysins are in progress. Endolysins (SAL200, CF-301, and P128) against *S. aureus* have reached the clinical trials (Table 2). Most of these studies involving endolysins are targeting different routes of infections caused by *S. aureus* [20].

## 12. Challenges in Clinical Trials

Despite the remarkable qualities, some challenges need to be addressed. In vitro studies are carried out using exponential phase bacterial cells where the endolysins have shown excellent efficiency. Studies have shown that stationary phase cells are less susceptible to endolysins due to peptidoglycan maturation [160]. Modifications occur during acetylation/deacetylation of glycan chains; peptide amidation is responsible for resistance against endolysins in both Gram-positive and Gram-negative bacterial cells [3].

Unwanted immune response to endolysins decreases their efficiency inside the human body or causes anaphylaxis [173]. Using traditional mouse models, studies reported the unreliable immunogenicity of endolysins in humans. During preclinical and clinical trials of SAL200, antibodies were produced in varying degrees in rats, dogs, monkeys, and humans [67]. More trials are needed to understand the immunogenic nature of endolysins. For endolysin bioengineering, X-ray crystallography is performed for the structural characterization of these enzymes. However, difficulty in crystal formation of endolysin is also a considerable challenge [68].

## 13. Conclusions and Future Perspectives

Due to the rise in multi-drug resistance bacterial infections across the globe, endolysins as a novel therapeutic approach has received significant attention. Endolysins are an attractive option as they show lytic potential against numerous bacterial species of concern within veterinary and human medicine and show advantages within the agricultural and biotechnology sector. Additionally, endolysins show promising advances against biofilm formation. Interestingly, the recent exploration of resistance, safety, immunogenicity, and the synergy with antibiotics has advanced the research of endolysins further.

Endolysins are the best alternative therapeutic approach to cure and treat multi-drug resistant bacteria. Up to now, many endolysins are reported which show good results in treating antibiotic-resistant bacteria. However, endolysin also has some challenges. Endolysins show good efficacy to treat Gram-positive bacteria, but due to Gram-negative bacterial outer membrane barrier, it shows less activity to treat Gram-negative bacteria [42,174]. Another limitation of endolysin is its short **in-vivo** half-life, due to the production of cytokines’ inflammatory response, and the neutralizing antibodies against it. Endolysin provokes an immune response when it is used systematically, so due to immune response, it loses its enzymatic lytic activity *in-vivo* [133,175,176].

New strategies are needed to develop universal chimeric lysin, to cross the Gram-negative bacterial outer membrane barrier, and to overcome these immunological responses against endolysin. While endolysins are proving to be advantageous as novel therapeutics, further research is required to consider their formulation and engineerability towards clinical trials.

## Figures and Tables

**Figure 1 antibiotics-10-00124-f001:**
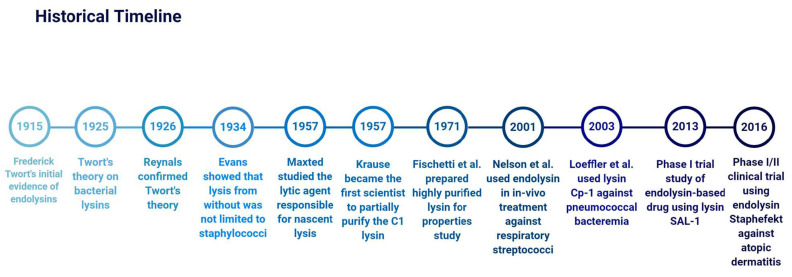
Timeline showing a brief history of endolysin research work.

**Figure 3 antibiotics-10-00124-f003:**
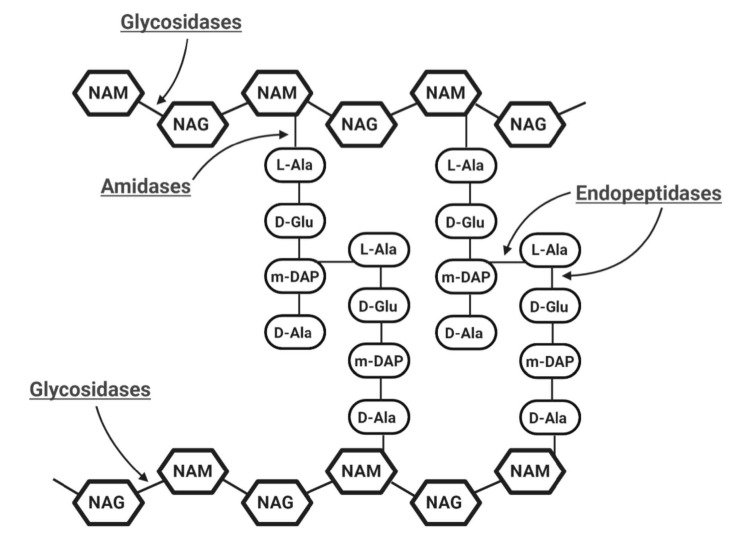
Schematic representation of the basic structure of bacterial cell wall peptidoglycan with possible catalytic activities of endolysins indicated. A sub-class of Glycosidases, N-acetyl-β-D-muramidases cleave the β-1,4 bonds between NAM (N-acetylmuramic acid) and NAG (N-acetylglucosamines), and N-acetyl-β-D-glucosidases cleave the β-1,4 bonds between NAG and NAM residues. N-acetylmuramoyl-L-alanine amidases which are amidases cleave the amide bonds between NAM and L-alanine. Endopeptidases cleave interpeptide and stem peptide–interpeptide bridges.

**Figure 4 antibiotics-10-00124-f004:**
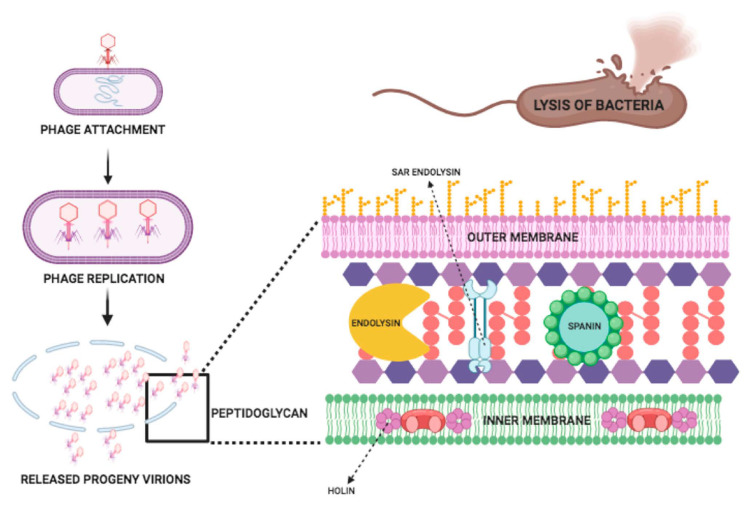
Mechanism of action of endolysin.

**Figure 5 antibiotics-10-00124-f005:**
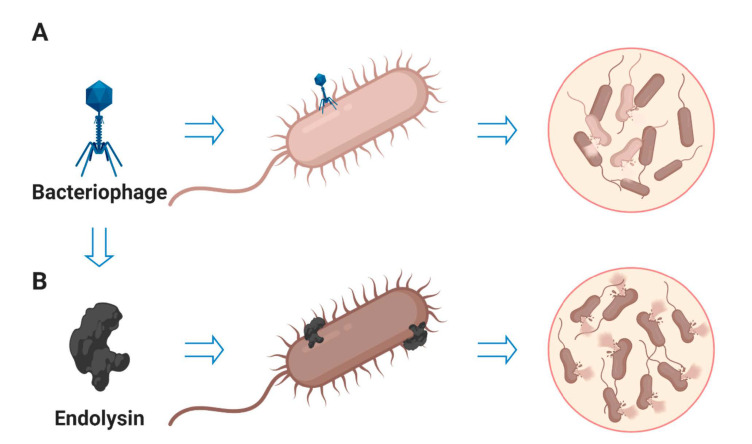
The schematic diagram shows the lytic efficacy of bacteriophages and endolysins. (**A**) Bacterial strains getting resistance against specific phage after phage therapy. (**B**) Phage endolysin therapy shows better killing efficiency than phage cocktail therapy.

**Figure 6 antibiotics-10-00124-f006:**
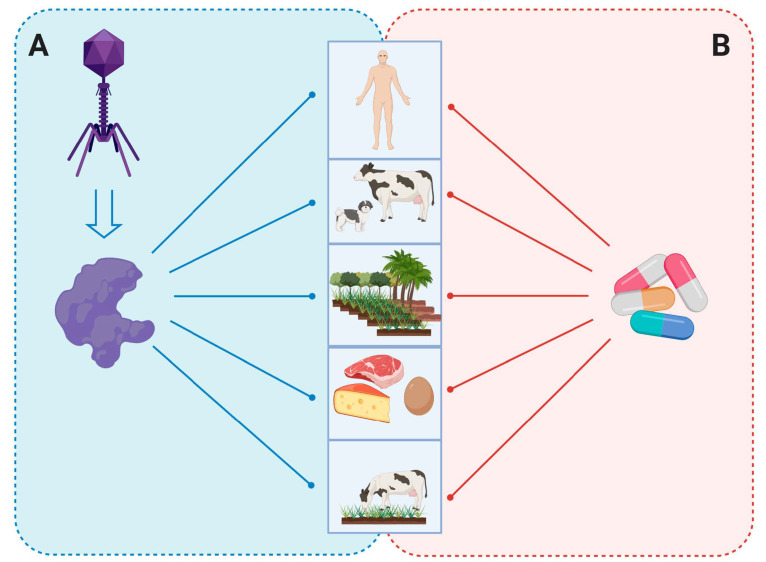
Bacteriophage endolysins as an antibacterial agent. (**A**) Phage-derived endolysin protects humans, animals, plants that are extremely infected with antibiotic-resistant pathogens and also inhibits the prevalence of antibiotic resistance via other food chains. (**B**) Overuse of antibiotics in food chains and human and veterinary medicine causes uncontrollable bacterial infections with multidrug resistance that leads to future pandemics.

**Figure 7 antibiotics-10-00124-f007:**
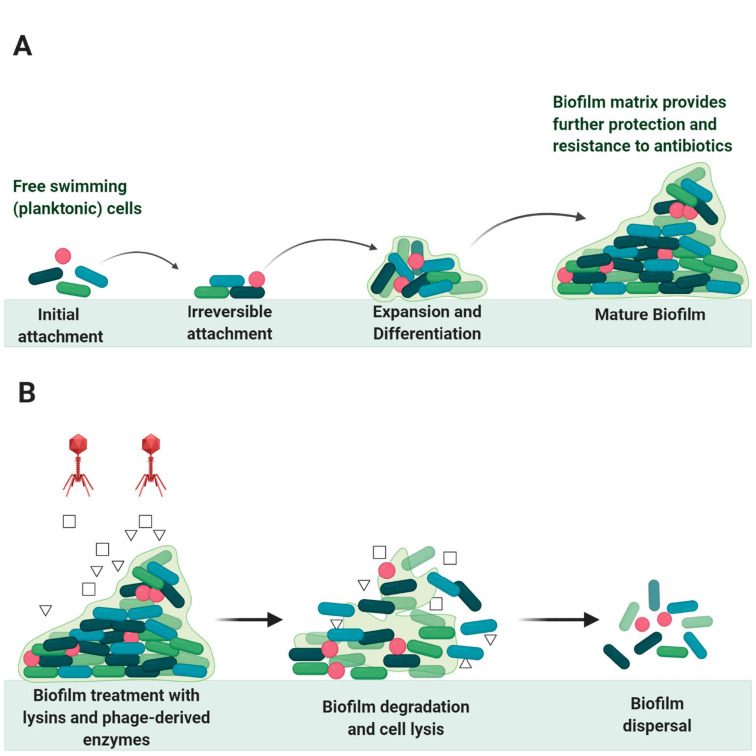
Biofilm growth stages and degradation: (**A**) the stages of bacterial biofilm formation from the initial attachment to mature biofilm formation, (**B**) the process of biofilm dispersal using phage endolysins and phage-derived enzymes as agents of biofilm degradation and bacterial cell lysis.

**Figure 8 antibiotics-10-00124-f008:**
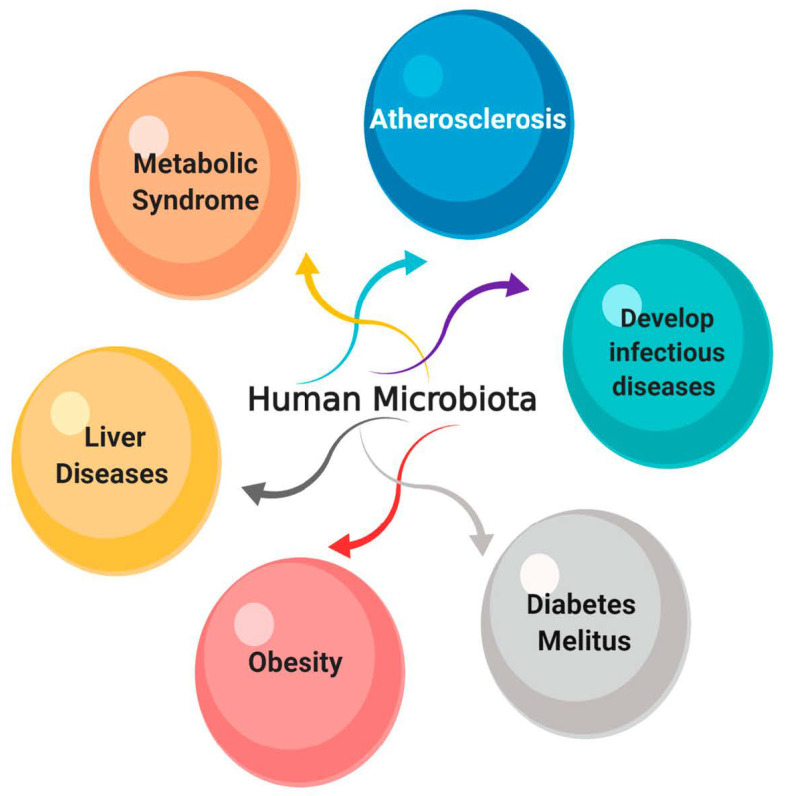
Relationship of microbiota disturbance with developing diseases in different systems.

**Table 2 antibiotics-10-00124-t002:** List of endolysins in human clinical trials.

Endolysin	Study	Phase	Clinical Trials Identifier	Status
**CF-301**	Safety, Efficacy, and Pharmacokinetics of CF-301 vs. Placebo in Addition to Antibacterial Therapy for Treatment of *S. aureus* Bacteremia	III	NCT04160468	Ongoing trial
**N-Rephasin® SAL200** **(Intron Biotechnology, Inc., South Korea)**	To evaluate safety and to explore the efficacy of a single intravenous dose of N-Rephasin® SAL200 (3 mg/kg)	IIa	NCT03089697	Ongoing trial
**P-128**	Safety & Efficacy of an Antibacterial Protein Molecule Applied Topically to the Nostrils of Volunteers and Patients	II	NCT01746654	Completed

## Data Availability

Not applicable.

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
