# Peer review of "Phage-Encoded Endolysins"

_antibiotics, 2021, doi:10.3390/antibiotics10020124_

Round 1

Reviewer 1 Report

Although I appreciate the authors effort to prepare the review I think that the manuscript has serious flaws and I can’t support it in a current state for publication in Antibiotics. I think that review needs to be a knowledge compendium allowing the reader to cite it without digging into the cited literature. Many aspects are too simplified and shown in a very cursory way. The authors write about holins, but forget about SAR endolysins, spanins and pinholins (see Ry Young review: Adv Virus Res. 2019;103:33-70. doi: 10.1016/bs.aivir.2018.09.003). Holins form micron-scale holes in the inner membrane through which the endolysins escape to the periplasm and degrade the peptidoglycan. This is not shown in Figure 4, which is very misleading and should not be propagated. The configuration models of “Gram-positive and Gram-negative endolysins” are also incorrect. First, catalytic domains in endolysins structures are called enzymatically active domains (EADs) and this nomenclature is correct. Second, there is no such term “Gram-negative/Gram-positive endolysin” because Gram staining distinguish bacteria, not endolysins. For possible domains configuration of endolysins see Hugo Oliveira review: J. Virol. 87, 4558–4570.http://doi.org/10.1128/JVI.03277-12). For example, some endolysins comprise more than one CBD, e.g. the streptococcal endolysin λSa2 has two CBDs. Some modular endolysins in Gram-negative infecting phages have two C-terminal EADs, etc. Next, MIC assay is only performed in liquid media, not on agar plates (line 214-215). This section is too superficial.

Monophage therapy is not routinely used today, phage coctails are used instead and these should be compared to endolysins efficacy; Figure 5.

Summarizing, I think that first part about endolysins structure, historical perspective and methods used to test them was written by one person and the other one wrote the part about clinical applications. The second part is much more interesting and better English is used as well. It is a bit out to date (see specific comments) but it can be amended. Especially sections 1, 2, 3, 4 and 5 need to be carefully rewritten. The section about distinction between N-acetyl-β-D-glucosaminidases, N-acetyl-β-D-muramidases, lytic transglycosylases, N-acetylmuramoyl-L-alanine amidases and endopeptidases needs to more clearly prepared.

If the authors will rewrite this sections I can support the manuscript for publication.

Specifically:

„… never reported to develop resistance..”, line 46 I would not make it so simple. I think it is worth mentioning that the resistance to other peptidoglycan hydrolases such as lysozyme or lysostaphin exists.

again not only holins regulate endolysin’s dependent peptidoglycan degradation, line 53

What do authors mean by writing “… short domains…” line 54-55; How many amino acids these domains have? The authors don’t mention cell wall binding domains (CBDs) but I think they should.

I don’t think that the scientific community was ignorant towards endolysins applicability, line 59

What about Prof. Fischetti work? Krause, a physician in the laboratory of Maclyn McCarty at the Rockefeller University was the first to partially purify the C1 lysin (see Viruses. 2018 Jun; 10(6): 310. doi: 10.3390/v10060310). Nothing is mention about endolysins purification attempts. The topic should be expanded. Maxted worked on filtrates and from the text the reader can understand that the author worked with purified lysin, line 86

Figure 1, nothing is mention about year 1971 and Prof. Fischetti work in the text.

In the nomenclature of endolysins the phrase catalytic domain (CD) is not used, instead enzymatically active domain (EAD) exist and is correct.

The nomenclature ”amidohydrolases” is not used among scientists working with endolysins. Simple word amidases is used.

CDs are very specific toward cell wall components …, line 196. From the text the reader can understand that the catalytic domain binds polysaccharides, but it is not true.

different font, line 209 – 210.

zymogram not zymo-gram, line 212

Cystic fibrosis is not a chronic bacterial infection, line 223

Holin itself is not a lytic enzyme, line 234, 236

De Corte et al., did not use endolysin LoGT-008 in their research, lines 292-294

lysi (?), line 483

Section 9; the authors mention Gram-positive bacteria and endolysins directed toward them, but don’t write about endolysins from Gram-negative background. These enzymes, in general, don’t have the CBD and often have broad host range.

S. aureus – for bacterial nomenclature use italics; line 668, 697

What the authors mean by writing: “..The globular structure comprises both catalytic domains.”

CF-301 is the first endolysin in a phase III clinical trial for the treatment of patients with S. aureus bloodstream infections, including right-sided infective endocarditis (ClinicalTrials.gov, NCT04160468). The Table 2 is not up to date.

Nothing is mentioned about novel VersaTile driven platform used to develop endolysins against Gram-negative bacteria (see Science Advances  03 Jun 2020: Vol. 6, no. 23, eaaz1136 doi: 10.1126/sciadv.aaz1136).

Author Response

Dear Reviewer,    Thanks for your valuable time and comments for the improvement of our submitted review article. We have made considerable changes as per the comments and suggestions made by you and hope they are as per your satisfaction and requirement.    The article has been considerably modified in the text as well as in the figures and point by point response is also provided for the same in the attached response to reviewers comments.   I hope these changes will be helpful in improving the quality of our review article pending major revision.     Best regards,   Fatma Abdelrahman 

Reviewer 2 Report

The current review manuscript entitled “Phage encoded endolysins” by Abdelrahman et. al., attempts to review the literature on phage endolysins, an important class of proteins with great potential for therapeutic use. The review covers the history, biochemical details of endolysin mode of action and classification, domain architectures, and applications in humans, animals, and plants. Although it is an interesting subject, overall, the manner in which the information is presented in this review makes the reader dazed and confused. A comprehensive re-write is needed to make this review less verbose and enjoyable to read.

Regulation of expression section needs to be rewritten or removed altogether. The authors attempt to provide a generalized description of late gene expression in phage lambda but fail to provide any meaningful conclusions that could be extrapolated to most phages. What about genomes that do not have Cro-like proteins? Regulation in virulent phages doesn’t require Cro.

Line 45: Instead of citing another review it would be beneficial to cite some primary literature that shows activity against Gram negative bacteria. The review cites examples of Endolysins working against Gram negative but the hosts have to be permeabilized with EDTA or the endolysins have cationic alpha helices that aid in uptake of the endolysins. Typical concentration of endolysins used in those studies is final concentration of 500 μg ml−1 or used in permeabilized bacteria. Moreover, the concentrations of 500 mg/L are probably not medically feasible for a protein-based therapeutic.

Line 45: “Being an essential component of the phage life cycle”,… Not essential in all phages. For example filamentous phages do not need Endolysins. It is better to say that it is essential in phages with lytic life cycle.

Line 52: Refer to the Figure. Since you already have a figure in the manuscript showing different types endolysins.

Line 54: It is a hole not a pore. Pore refers to a small opening but some holin holes are massive >100 nm in diameter.

Lines 57-59: Too long and confusing. Fragment and rephrase. How can unavailability of data cause hot debate?

How can the following cause hot debate?

  1. unavailability of sufficient clinical data
  2. lack of in vivo studies
  3. Ignorance of people.

I presume it would be a rather short debate ending with phrase like "get more data".

Lines 60-62: I disagree. Bacteria have evolved barriers such as outer membrane, exopolysaccharides etc to protect themselves against free floating endolysins.

Line 73: “VAPGHs are secreted at the initial stage” released from tail tip NOT secreted.

Line 74: “Lysation” is NOT a real word.  “ in vitro lysis of bacteria”

Line 89: To be consistent, replace lysozyme with muramindase. I understand that the authors are quoting the sentence but it help readers if there was a sentence immediately following this that clarified lysozyme is a muramindase.

Line 115 and throughout the text: F in figure was capitalized for Figure 1 but not for Figure 2.

Line 121: There are also inner membrane tethered endolysins in Gram negative phages. Famous example is the lyz of phage P1 which is a SAR-Endolysin.

Line 145: Add (f) SAR endolysins.

Lines154-155: Although this appears to be a logical explanation but the authors have either ignored or unaware of the outer membrane disrupting proteins (spanins) in Gram negative phages. In the presence of spanins the endolysins would be released into the environment. However, the presence of OM barrier in the neighboring cells would prevent free-floating endolysins from gaining access to the peptidoglycan.

Lines 156-157: Please re-word this. In the current format it suggests that presence of mycolate layer magically introduces two forms of CD lysins.  Maybe something like this “To counteract or overcome the outer thick mycolate layer linked to the peptidoglycan layer in mycobacteria, the mycobacteria phages have evolved two forms of CD lysins”.

Line 173: “and in less than 30 minutes a life cycle” some cases not all the time.

Line 177: “pore a hole in”  redundant and confusing.

Lines 179-181: This applies to Gram negative because of the OM but not for Gram positive bacteria. And the reference for this section is a review from nearly 20 years ago. I am sure there is more known about lysis in the past 20 years.

Lines182-186: This section is redundant. Was covered in the preceding section.

Lines 200-201: Please re-write the sentence in a more direct way.

Lines 203-204: holes in the cell wall or holes in the inner membrane?

Lines 207-208: I guess the authors are trying to say that "Only one enzyme will be enough to cleave sufficient PG bonds to osmolytic death of the bacteria".

Lines 220-225: Why are bacteriophages introduced here? Previous sections mentioned bacteriophages but didn't explain what they were. If the intended audience of this review are people familiar with bacteriophages and endolysins then this really simplistic explanation is not needed. One would assume that readers would already be familiar with the terms.

Lines 228-229: I do not understand. How can mere attachment of phage regulate the expression of holins.

Lines 233-234: Holins are NOT enzymes. At least, not in this universe.

Line 333: encounter?? Or counteract?

Line345: “antimicrobial with 10 hundred folds compare” does it mean 1000 fold??

Author Response

(The authors gave the same response as above.)

Round 2

Reviewer 1 Report

The manuscript ‘Phage-encoded endolysins’ is partially improved but still requires several corrections. See specific comments below:

proteins, line 52

something is incorrect in these sentences: “…likely due to the horizontal gene transfer and among phage-host systems over a long time  [1]. However, bacterial hosts do develop resistance mechanisms against free endolysins include the presence of the outer membrane (capsule) and exopolysaccharides (biofilms).”, lines 53-56 Please, check the language once again throughout the manuscript.

What the authors mean by writing this sentence: “However, the lack of in vivo studies and the subsequent shortfall in sufficient clinical trials has prolonged the use of bacteriophages and endolysins (phage therapy).” lines 62-64 I think that “prolonged” is not a proper word here.

Gram-negative, line 211

I would improve Figure 4. In this Figure it is not shown that holins make small pores in the inner membrane. Are the authors sure that endolysin directly binds to holin and they make a complex?

binds not bind, line 121

I think that: “Glycosidases include the N-acetyl-β-D-muramidases, cleaving β-1,4 bonds between MurNAc and GlcNAc; N-acetyl-β-D-glucosamidases, cleaving β-1,4 bonds between GlcNAc and MurNAc residues;….” is too complicated. Please, rewrite it make it more simple.

I think it is better to write: cleave, catalyze etc. not cleaving, catalyzing, lines 138, 148, 151

insert (VANY) after  L-alanoyl-D-glutamate endopeptidase, line 153, include shortcuts where possible.

no CWBD but CBD, legend to Figure 2

‘a Model’ change for ‘a model’, legend to Figure 2

citation at the end of the sequence, line 164

not ‘is consisting’ of but ‘consists of’, line 191

not …bacteriophage, their lytic enzyme (endolysins) has also received significant attention…. but ….bacteriophages, their lytic enzymes (endolysins) have also received significant attention, particularly for…. lines 274, 275

“Particularly, food animals such as cattle, poultry, and swine have been noted as a significant threat to the public health crisis” the meaning of this sentence is misleading, line 337

I think that original papers of prof. Briers should also be cited there, lines 602-609

in vivo and in vitro in italics, line 612

Author Response

Dear Reviewer,    Thanks for your valuable time and comments for the improvement of our submitted review article. We have made considerable changes as per the comments and suggestions made by you and hope they are as per your satisfaction and requirement.    The article has been considerably modified in the text as well as in the figures and point by point response is also provided for the same in the attached response to reviewers comments.   I hope these changes will be helpful in improving the quality of our review article pending major revisions.   Best regards,   Fatma Abdelrahman

Reviewer 2 Report

The authors have adequately addressed the issues that I had raised. However, some of the graphical elements appear to have been adopted from Biorender.com. Please acknowledge the source.  

Author Response

(The authors gave the same response as above.)
